

# Is publication bias present in gastroenterological research? An analysis of abstracts presented at an annual congress

Chase Meyer, Kaleb Fuller, Jared Scott and Matt Vassar

Oklahoma State University College of Osteopathic Medicine, Tulsa, OK, United States of America

## ABSTRACT

**Background**. Publication bias is the tendency of investigators, reviewers, and editors to submit or accept manuscripts for publication based on their direction or strength of findings. In this study, we investigated if publication bias was present in gastroenterological research by evaluating abstracts at Americas Hepato-Pancreato-Biliary Congresses from 2011 to 2013.

**Methods**. We searched Google, Google Scholar, and PubMed to locate the published reports of research described in these abstracts. If a publication was not found, a second investigator searched to verify nonpublication. If abstract publication status remained undetermined, authors were contacted regarding reasons for nonpublication. For articles reaching publication, the $P$ value, study design, time to publication, citation count, and journals in which the published report appeared were recorded.

**Results**. Our study found that of 569 abstracts presented, 297 (52.2%) reported a $P$ value. Of these, 254 (85.5%) contained $P$ values supporting statistical significance. The abstracts reporting a statistically significant outcome were twice as likely to reach publication than abstracts with no significant findings (OR 2.10, 95% CI [1.06–4.14]). Overall, 243 (42.7%) abstracts reached publication. The mean time to publication was 14 months and a median time of nine months.

**Conclusion**. In conclusion, we found evidence for publication bias in gastroenterological research. Abstracts with significant $P$ values had a higher probability of reaching publication. More than half of abstracts presented from 2011 to 2013 failed to reach publication. Readers should take these findings into consideration when reviewing medical literature.

## INTRODUCTION

The practice of evidence-based medicine integrates clinical expertise with the best available clinical research evidence (*Sackett et al., 1996*). This movement promotes the use of high-quality clinical research in clinical decision making (*Masic, Miokovic & Muhamedagic, 2008*). If treatment decisions are to be truly evidence based, it is necessary that the literature accurately reflect an intervention's effectiveness (*Dickersin, 1990*). This information is thwarted, however, when research regarding efficacy does not reach publication.

Corresponding author
Chase Meyer,
chase.meyer@okstate.edu

Publication bias is one such reason why studies fail to be published. Publication bias is the tendency of investigators, reviewers, and editors to submit or accept manuscripts for publication based on their direction or strength of findings (*Dickersin, 1990*). A substantial body of evidence supports that studies reporting negative (nonsignificant) findings are less likely to be published (*Fanelli, 2010*; *Easterbrook et al., 1991*). The implications of publication bias are far reaching for clinical decision making owing to the possibility of overestimated treatment effects. For example, several meta-analyses have re-evaluated the efficacy and safety of antidepressants and determined that their therapeutic value has been overestimated when considering data used from unpublished studies (*Turner et al., 2008*; *Eyding et al., 2010*; *Barbui, Furukawa & Cipriani, 2008*; *Whittington et al., 2004*). Interestingly, while publication bias has been widely discussed in many areas of medicine such as cancer (*Harris et al., 2010*; *Saeed et al., 2011*; *Paulson et al., 2011*; *Salami & Alkayed, 2013*; *Sartor, Peterson & Woolf, 2003*; *Takeda et al., 2008*) and anesthesiology (*Chong et al., 2016*; *De Oliveira Jr et al., 2012*; *Lim et al., 2016*; *Sukhal et al., 2017*; *Jones, 2016*; *Hedin et al., 2016*), this important issue has received limited attention in gastroenterology with mixed results (*Timmer et al., 2002*; *Shaheen et al., 2000*; *Eloubeidi, Wade & Provenzale, 2001*). One example of the implications of publication bias concluded that the incidence of Barrett's esophagus may be overestimated due to publication bias (*Shaheen et al., 2000*). If the incidence is lower than previously assumed, there might be an overemphasis on the benefits of costly screening programs, leading to a loss of resources.

Publication bias may also arise during scientific meetings. Organizations hold scientific meetings to allow researchers to come together to discuss new and ongoing topics related to their field of interest through oral and poster presentations of original abstracts. Because of the competitive publication process, abstracts that are presented likely represent strong research that may influence the current literature (*Frank et al., 2017*). However, not all presentations will be published, and nonpublication of findings may be harmful to patients, result in unnecessary duplication of efforts, contribute to research waste, and prevent results from being included in systematic reviews (*Durinka et al., 2016*).

In this study, we measured the publication rate of abstracts presented at Americas Hepato-Pancreato-Biliary Association (AHPBA) Congresses and established whether publication bias may have occurred between abstract presentation and publication. We also evaluated the length of time to publication and which journals most frequently publish AHPBA abstracts. For unpublished abstracts, we contacted authors to determine the reason for nonpublication.

## METHODS

### Oversight and reporting
This study did not meet the regulatory definition of human subjects research as defined in 45 CFR 46.102(d) and (f) of the Department of Health and Human Services' Code of Federal Regulations and therefore was not subject to Institutional Review Board oversight. We applied relevant Statistical Analyses and Methods in the Published Literature guidelines for reporting descriptive statistics.

### Locating conference abstracts

We located the AHPBA abstracts from 2011 to 2013 through the AHPBA website (*AHPBA, 2017*). We selected this time period based on previous literature describing the need to allow adequate time for a conference abstract to be published (*Durinka et al., 2016*). After locating the AHPBA abstracts we began the search process.

### Search strategy for published manuscripts of conference abstracts

Using a predefined search algorithm, we attempted to locate the published report of conference abstracts (Fig. 1). The search algorithm was developed by two investigators (JS and MV) and pilot tested on 25 abstracts. We assessed the optimal order in which to search databases (for example, Google first, Google Scholar second, and PubMed third). We also varied the searches by using combinations of keywords and author names and used full title searches to determine which strategy would most precisely locate published reports. Ultimately, our search was completed using three databases: PubMed, Google Scholar, and Google. One investigator (CM) first searched these three databases using the full conference abstract title. If this strategy failed, this investigator performed searches using an author's last name and keywords from the abstract. When CM could not locate a published report, a second investigator (KF) repeated the search strategy.

If the second investigator (KF) could not locate the published report, CM sent a standardized email to an author of the conference abstract (see Appendix 1). This email gave authors the opportunity to comment whether the study had reached publication and provide the reference for the publication. If the author indicated the report was not published, abstract authors were asked to provide a reason for nonpublication. Our standardized response options for nonpublication were based on a systematic review by *Song, Loke & Hooper (2014)* which analyzed 38 survey reports on investigator-reported reasons for nonpublication.

For studies that were not found to be published and also contained negative findings, we also searched for them on Faculty of 1000 (F1000), BioMed Central (which includes the archives of the Journal of Negative Results in Biomedicine), and Cureus, as these sources publish studies with negative findings.

### Data collection

Our search dates ranged from June 13, 2017 to June 20, 2017. Once a published study that was thought to be a conference abstract was located, we compared the author list, methods, and results between them. If at least two of the following criteria were met, we considered the abstract published: (1) results in both reports matched; (2) the methodology was similar; and (3) the first author of the conference abstract was included in the author list of the published study.

Data were extracted from the published studies by CM using a Google form. The following information was extracted: publication title, institution of first author, date submitted to journal (when available), date accepted for publication (when available), date of in print publication (when available), date of online publication (when available), sample size (when available), journal name, number of citations, and whether there was

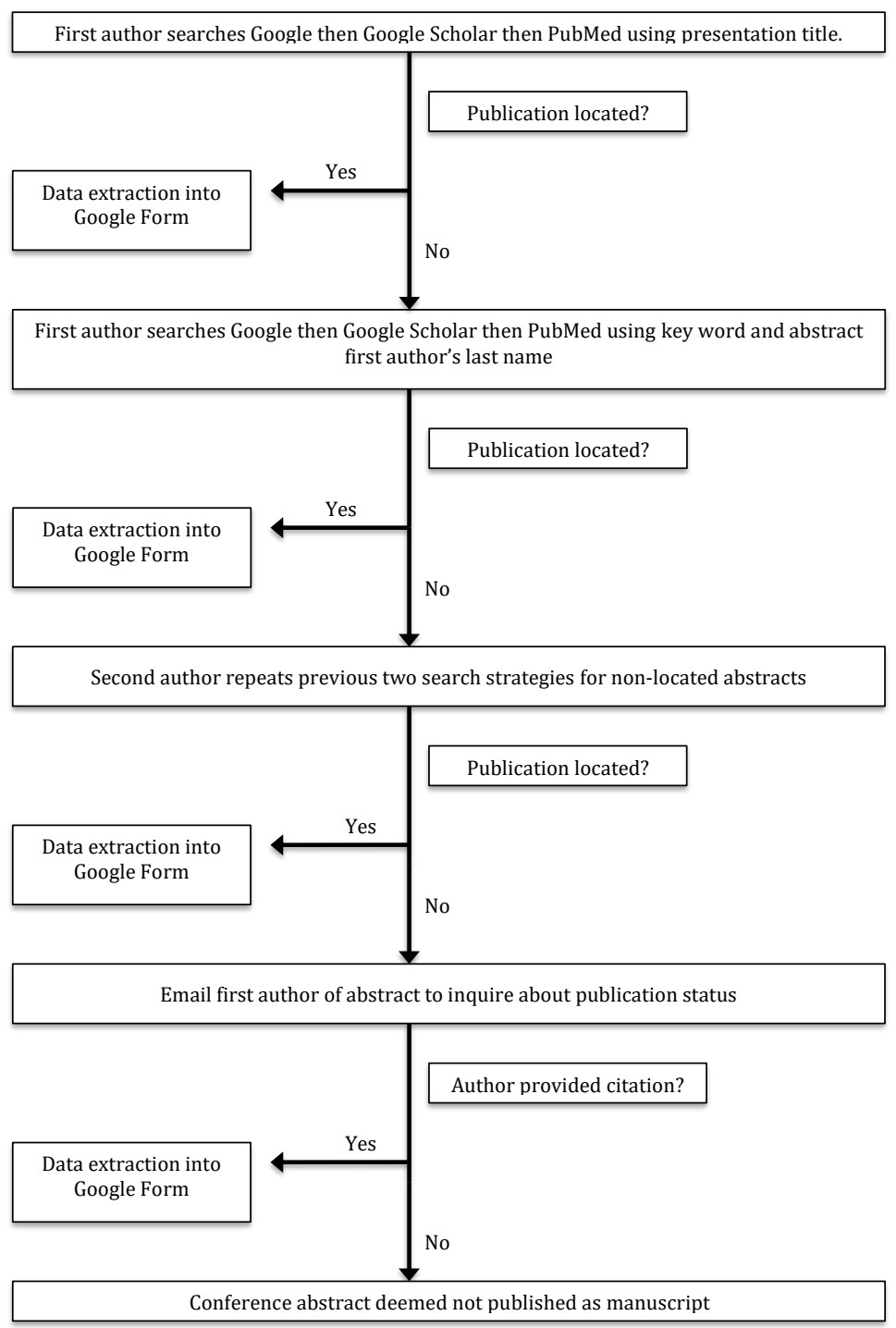

**Figure 1  Flow diagram for locating articles.**

a significant outcome ($P < 0.05$). The time to publication was calculated based on the number of months between the first date of the conference and the publication date in print or online, whichever occurred first. Descriptive statistics are reported as both means and medians. The reporting of means allows for interpretation of our findings in the context of other studies. Medians are reported due to the non-normal nature of the time to publication and citation count variables. A Mann Whitney U test was used to evaluate for differences on citation count between studies with positive and negative results. For this analysis, we included sample size and study design as potential predictors of publication. Logistic regression was used to evaluate the associations of sample size and study design with publication. We classified abstracts according to study design, which included such designs as: cohort studies, case studies, and randomized controlled trials. For study design, the investigators retained all study designs that represented at least 10% of abstract presentations for stability of the regression coefficients. Data analyses were performed using Microsoft Excel and Stata 13.1.

## RESULTS

A total of 12 abstracts were found to be published before presentation and were therefore excluded from this study. A *P* value was reported in 297 (52.2%) of 569 abstract presentations. Of those with a reported *P* value, 254 (85.5%) reported significant outcomes. Of the 254 that reported a significant outcome, 139 (54.7%) went on to reach publication. Of the 41 abstracts that reported negative outcomes, 15 (36.6%) went on to reach publication. No abstracts that reported only negative outcomes were found to be published on F1000, BioMed Central, or Cureus. Abstracts with at least one significant outcome were twice as likely to reach publication than abstracts with no significant findings (OR 2.10; 95% CI [1.06–4.14]. The most common study design was retrospective analysis, with 313 abstracts. Of these, 150 (47.9%) reached publication. The least common study design was the randomized controlled trial with three abstracts; however, all reached publication. Full study design results can be found in Table 1.

From 2011 to 2013 there were 569 abstract presentations, of which 243 (42.7%) reached publication (Fig. 2). The mean time to publication was 14.0 months and a median time of nine months. For 2011, 79 of 168 (47%) abstracts reached publication, with a mean time of 12.5 months and a median time of seven months. For 2012, 89 of 201 (44.3%) abstracts reached publication, with a mean time of 14.8 months and a median time of 12 months. For 2013, 75 of 200 (37.5%) abstracts reached publication, with a mean time of 14.5 months and a median time of 11 months. A graph displaying cumulative time to publication is presented in Fig. 3. For abstracts with significant findings, the mean time to publication was 12.1 months and a median time of eight months. For abstracts with negative findings, the mean time to publication was 13.6 months and a median time of eight months.

Citation counts were not significantly different between studies with negative results (Median = 12, Interquartile range (IQR) 8–28) and studies with positive results (Median = 13, IQR 3–74; $Z = -.34$, $P > .05$). Results from logistic regression indicated that sample size (OR 1.00, 95% CI [.99–1.00]) retrospective study design (OR .90, 95% CI [.46–1.76]),

**Table 1  Study design of abstracts and the number that reached publication.**

| Study design | Abstracts by study type ($n$) | Abstracts reporting a significant $P$ value by study type ($n$) | Abstracts reaching publication by study type ($n$) |
|---|---|---|---|
| Randomized controlled trials | 3 | 2 | 3 (100%) |
| Cohort | 17 | 8 | 11 (64.7%) |
| Retrospective analysis | 313 | 183 | 150 (47.9%) |
| Case report | 58 | 1 | 12 (20.7%) |
| Video report | 14 | 0 | 3 (21.4%) |
| Survey report | 8 | 5 | 2 (25%) |
| Basic science | 29 | 15 | 11 (37.9%) |
| Single assignment | 19 | 5 | 11 (57.9%) |
| Systematic review/meta-analysis | 7 | 3 | 2 (86.5%) |
| Prospective analysis | 40 | 24 | 23 (57.5%) |
| Animal study | 16 | 8 | 8 (50%) |
| Cost analysis | 3 | 0 | 2 (66.7%) |
| Technique report | 42 | 0 | 5 (11.9%) |

or case reports (OR 4.70, 95% CI [.61–36.35]) were not predictive factors of publication status (model $\chi^2 = 4.91$, $p > .05$). Other study designs were encountered too infrequently to include in this analysis.

Fifty-nine journals published abstracts presented at the AHPBA congresses from 2011 to 2013 (Table 2). There were eight journals that published five or more full-text articles from the abstract presentations, accounting for 70% of publications. The most frequent journal in which abstracts were published was the conference's own journal, HPB, with 115 of the 243 publications (47.3%). Other notable journals included *Annals of Surgical Oncology* 14 of 243 (5.7%) and the *Journal of Gastrointestinal Surgery* 12 of 243 (4.9%). Table 2 also includes data on the distribution of positive and negative results published in these journals.

Some 326 abstract presentations could not be found published as full papers. Email addresses were obtained for 298 authors of these abstracts. Twenty-eight authors of the abstract presentations could not be associated with an email address. Additionally, 42 emails were returned as invalid addresses. Thirty-four authors (34/256, 13.3%) responded to emails. Of these 34, 10 authors provided information that their presentation was published, while 24 reported that it never reached publication. The most common reasons for not reaching publication were lack of time (7), lack of manpower (4), in preparation or under review (4), and results negative or not important (3).

## DISCUSSION

Our study, like others before it, revealed that investigations with at least one statistically significant outcome had a higher probability of reaching publication than those with insignificant or null findings (*Hopewell et al., 2009*; *Tang et al., 2014*). Furthermore, our

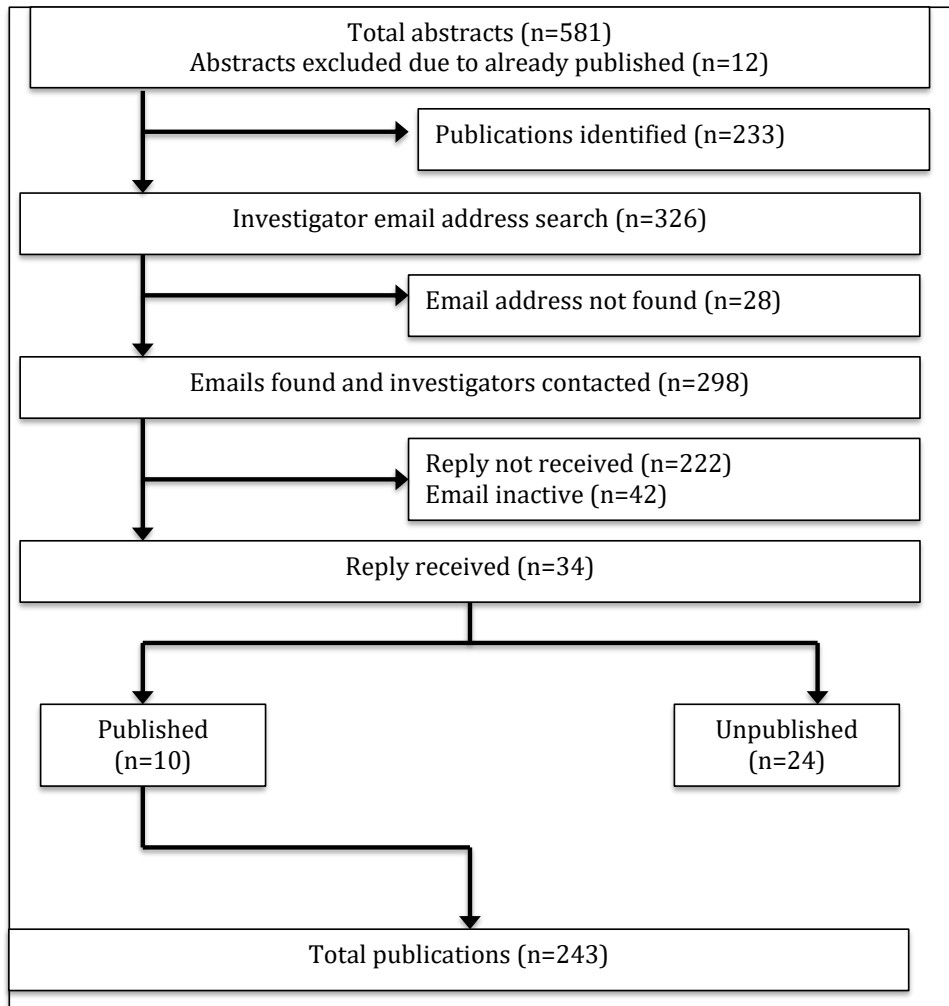

**Figure 2  Flow diagram outlining search results.**

findings on study design are consistent with *Ball, Dixon & Vollmer (2016)* for abstracts presented from 2005 to 2015 at the AHPBA.

It could cogently be argued that *P* values are misused in the medical literature. For example, *Amrhein, Korner-Nievergelt & Roth (2017)* reminds us that the dichotomization of outcomes into significant and non-significant may contribute to irreproducibility, and that data dredging, p-hacking, and publication bias should be addressed by the elimination of significance thresholds. The majority of abstracts in our sample were retrospective in design—not adequately powered, well-conducted randomized trials. This misuse has been attributed to the mistaken understanding that *P* values are, "simple, reliable, and objective triage tools for separating the true and important from the untrue or unimportant" (*Mark, Lee & Harrell Jr, 2016*). Alternatives to *P* value reporting are increasing in popularity. Reporting the effect size (for interpretation of clinical significance) and its confidence interval (for interpretation of the precision of the effect estimate) are advocated by the

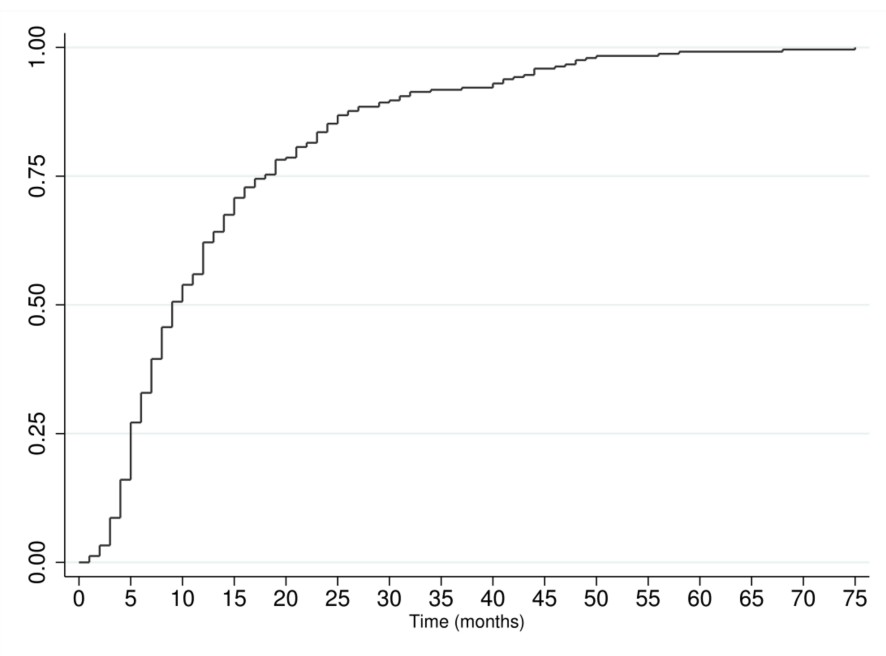

**Figure 3 Cumulative rate of publication.**

American Statistical Association and many journals (*Piccirillo, 2016*). Hence, the study design should carefully be considered when determining the appropriateness of calculating and reporting *P* values. The American Statistical Association's recent position statement on *P* values (*Wasserstein & Lazar, 2016*) is an excellent starting point for understanding issues related to the misuse of *P* values.

Furthermore, results of our study could be attributed, in part, to the misapplication of statistical analysis to underpowered studies. Publication bias, selective data analysis, and selective reporting of outcomes are more likely to affect underpowered studies (*Button et al., 2013*). Additionally, editors and reviewers may be more likely to reject underpowered, negative studies owing to their perception of being inconclusive or uninformative (*Evangelos et al., 2012*).

Finally, a large body of evidence has focused on the consequences of publication bias pertaining to randomized trials. In a systematic review of such trials, the omission of unpublished results may alter pooled effect estimates. Oftentimes, the pooled effect estimate is altered to make the intervention appear more favorable. In our study, the most common study design was a retrospective study. The effects of publication bias on these studies is still important, as epidemiological outcomes, such as prevalence and incidence, may be misestimated and correlational analyses may indicate inaccurate associations between clinical variables.

The publication rate of abstracts presented at the AHPBA from 2011 to 2013 was 42.7%. This rate is higher than what was found for the 2007 to 2009 congresses (33.4%) (*Durinka et al., 2016*). The rate of 42.7% is similar to the 44.5% rate reported by

**Table 2  Journals responsible for publishing abstracts.**

| Journal | Abstracts published (n) | Abstracts with significant outcome (n) | Abstracts with negative outcome (n) | Abstracts with no reported P value reaching publication (n) |
|---|---|---|---|---|
| American Journal of Transplantation | 1 | 1 | 0 | 0 |
| Annals of Surgery | 5 | 4 | 0 | 1 |
| Annals of Surgical Oncology | 14 | 8 | 0 | 6 |
| Annals of Vascular surgery | 1 | 1 | 0 | 0 |
| Archives of Surgery | 1 | 1 | 0 | 0 |
| Arquvois Brasileiros De Cirurgia Digestiva | 1 | 0 | 0 | 1 |
| BMC Cancer | 2 | 1 | 0 | 1 |
| British Journal of Surgery | 1 | 1 | 0 | 0 |
| Canadian Journal of Surgery | 1 | 1 | 0 | 0 |
| Cancer | 1 | 0 | 0 | 1 |
| Cancer Biology & Therapy | 1 | 0 | 0 | 1 |
| Cancer Investigation | 1 | 1 | 0 | 0 |
| Clinical Transplantation | 2 | 1 | 0 | 1 |
| Diagnostics | 1 | 1 | 0 | 0 |
| Endoscopy | 1 | 0 | 0 | 1 |
| European Journal of Surgical Oncology | 2 | 1 | 0 | 1 |
| European Journal of Trauma and Emergency Surgery | 1 | 0 | 1 | 0 |
| European Surgical Research | 1 | 0 | 0 | 1 |
| Genetics in Medicine | 1 | 0 | 1 | 0 |
| Hepatobiliary & Pancreatic Diseases International | 2 | 0 | 0 | 2 |
| Hepatogastroenterology | 2 | 1 | 0 | 1 |
| HPB | 115 | 74 | 7 | 34 |
| International Journal of Hyperthermia | 1 | 0 | 0 | 1 |
| International Journal of Surgery | 2 | 2 | 0 | 0 |
| International Journal of Surgical Oncology | 2 | 1 | 1 | 0 |
| JAMA Surgery | 1 | 1 | 0 | 0 |
| Journal of Biomedical Materials Research | 1 | 0 | 0 | 1 |
| Journal of Clinical Investigation | 1 | 0 | 0 | 1 |
| Journal of Gastrointestinal Cancer | 1 | 1 | 0 | 0 |
| Journal of Gastrointestinal Surgery | 12 | 8 | 0 | 4 |
| Journal of Hepato-biliary-pancreatic Sciences | 1 | 0 | 0 | 1 |
| Journal of Liver: Disease and Transplantation | 1 | 0 | 0 | 1 |
| Journal of Laparoendoscopic & Advanced Surgical Techniques | 2 | 0 | 0 | 2 |
| Journal of microwave surgery | 1 | 0 | 0 | 1 |
| Journal of Robotic surgery | 1 | 0 | 0 | 1 |
| Journal of Surgical Oncology | 7 | 5 | 0 | 2 |
| Journal of Surgical Research | 2 | 0 | 1 | 1 |
| Journal of Surgical Resection | 2 | 1 | 0 | 1 |
| Journal of the American College of Surgeons | 6 | 4 | 0 | 2 |
| Journal of the Medical Association of Thailand | 1 | 0 | 0 | 1 |

**Table 2** (*continued*)

| Journal | Abstracts published (*n*) | Abstracts with significant outcome (*n*) | Abstracts with negative outcome (*n*) | Abstracts with no reported *P* value reaching publication (*n*) |
|---|---|---|---|---|
| Journal of the Society of Laparoendoscopic Surgeons | 3 | 0 | 0 | 3 |
| Langenbeck's Archives of Surgery | 1 | 1 | 0 | 0 |
| Liver Transplantation | 1 | 1 | 0 | 0 |
| Molecular Oncology | 1 | 0 | 0 | 1 |
| Neoplasia | 1 | 1 | 0 | 0 |
| Open Journal of Organ Transplant Surgery | 1 | 0 | 0 | 1 |
| Pediatric Transplantation | 1 | 0 | 1 | 0 |
| PLOS ONE | 2 | 0 | 1 | 1 |
| Radiology Society of North America | 1 | 1 | 0 | 0 |
| Seminars in Liver Disease | 1 | 1 | 0 | 0 |
| Surgery | 5 | 3 | 0 | 2 |
| Surgical Endoscopy | 6 | 3 | 0 | 3 |
| Surgical Innovation | 2 | 1 | 0 | 1 |
| The American Journal of Surgery | 4 | 3 | 0 | 1 |
| The American Surgeon | 2 | 0 | 1 | 1 |
| Transplantation Proceedings | 2 | 1 | 1 | 0 |
| World Journal of Gastroenterology | 3 | 1 | 0 | 2 |
| World Journal of Gastrointestinal Pathophysiology | 1 | 0 | 0 | 1 |
| World Journal of Surgical Oncology | 2 | 2 | 0 | 0 |

(*Scherer, Langenberg & Von Elm, 2007*) in a systematic review of publication rates for 79 different biomedical conferences. In the context of other gastroenterology conferences, the 42.7% rate of publication was not the lowest. *Prendergast et al. (2013)* found that the British Society of Gastroenterology (BSG) had a rate of 17.4% in 2005 and *Raju et al. (2017)* found a rate of 30.9% for abstracts at the United European Gastroenterology week (UEGW). Reasons for not reaching publication were consistent with previous studies and most commonly pertained to lack of time, manpower, or negative results (*Pierson, 2004*; *Scherer et al., 2015*). Another consideration is that unpublished research may have never been submitted to a journal for review. Implications for not publishing are far reaching. With over $240 billion spent on health research each year, it is ideal to publish research to avoid unnecessary duplication, make data accessible, and prioritize future research (*Wolfenden et al., 2015*). Additionally, not publishing has been shown to decrease the likelihood of future patient volunteers (*Jones et al., 2016*).

The length of time to publication in our investigation (mean: 14 months, median: nine months) is favorable in comparison with BSG's mean time of 18.6 months and the same as the mean 14 months reported by Durinka et al. from 2007 to 2009 (*Durinka et al., 2016*; *Prendergast et al., 2013*). Furthermore, the AHPBA's median time to publication compares favorably to abstracts presented at other gastroenterology conferences (*Raju et al., 2017*; *Timmer et al., 2002*). A Cochrane review found that the length of time for publication can be influenced by publication bias (*Hopewell et al., 2007*). This review found that positive results were more likely to be published more quickly than those with

null or negative results. Studies published earlier are made available to clinicians sooner and therefore have clinical important implications to patient care. For example, the quicker publication of positive findings may result in systematic reviews overestimating treatment effects (*Scherer, Langenberg & Von Elm, 2007*). Furthermore, publication bias led to the widespread promotion of oseltamivir during pandemic seasons in 2005 and 2009 (*Gupta, Meenu & Mohan, 2015*). A systematic review highlighted this high risk of reporting and publication bias in trials assessing oseltamivir, finding limited evidence for its effectiveness in reducing symptoms, data unable to assess its effects on complications or transmission, and an increase in adverse side effects (*Jefferson et al., 2014*).

Publication bias is a cause for concern in the medical literature, as studies are often published due to the large magnitude effect sizes reported by investigators. Such effects are not likely reproducible in subsequent studies (*Baker & Dolgin, 2017*). Furthermore, systematic review efforts are often hindered when only data from published studies are available for inclusion in estimating summary effect estimates. For these reasons, it is important that action be taken to limit publication bias in the gastroenterology literature. To accomplish this aim, we propose that the following steps be taken:

1. Gastroenterology journals should pilot test, and work toward the adoption of, a two-stage peer review process in which the first stage is to evaluate the study on the methodological rigor of the study design before the outcomes of the study are known.
2. Gastroenterology journals and conferences need to place value on null and negative findings, encourage authors to submit their research regardless of the nature or direction of their findings.
3. Gastroenterology journals should consider including a negative results section of their journals as has been done in other medical fields (*Dirnagl & Lauritzen, 2010*).

We note the following limitations. First, while extensive measures were taken to determine the publication status of each abstract, it is possible that some were missed which could affect the results of this study. Changes in authorship, such as adding additional authors or rearranging the authorship order make matching abstracts with the published report more difficult. Changes to the title poses the same challenge. It is also possible that authors submitted interim results to the Congress yet published the final results, leading to incongruent aggregate outcomes and sample sizes. Other complicating factors include two abstracts being combined to form a single publication or a single abstract being parsed into multiple publications. Additionally, by excluding abstracts that did not report a *P* value from certain analyses could result in a bias in the reported estimate of incidence of significant results included in the studied publications. However, we feel confident that the abstracts identified as published by our search strategy are truly published versions of abstracts presented from 2011 to 2013 at the AHPBA annual congresses. While every effort was made to find author email addresses and make contact, we were unable to find a valid email address for authors of 70 presentations. Researchers only used Google, Google Scholar, and PubMed to find publications. Studies indexed in other databases that do not connect via Google may have been missed. However, given the exhaustive search and the efforts to email the authors, the number of omissions is likely very small. Given that 84% of abstracts were published in less than 24 months, our search interval was likely

adequate. Therefore, we caution readers that our findings should be considered a lower bound estimate of the publication rate.

## CONCLUSION

In conclusion, we found publication bias in the field of gastroenterological research. Abstracts with significant $P$ values were more frequently published than those with negative results. In addition, more than half of abstracts presented at the 2011 to 2013 AHPBA conferences failed to reach publication. Readers should take these finding into consideration when reviewing medical literature.

### Funding
The authors recieved no funding for this work.

### Competing Interests
The authors declare there are no competing interests.

### Author Contributions
- Chase Meyer performed the experiments, analyzed the data, contributed reagents/materials/analysis tools, prepared figures and/or tables, authored or reviewed drafts of the paper, approved the final draft.
- Kaleb Fuller performed the experiments, analyzed the data, contributed reagents/materials/analysis tools, authored or reviewed drafts of the paper, approved the final draft.
- Jared Scott conceived and designed the experiments, contributed reagents/materials/analysis tools, authored or reviewed drafts of the paper, approved the final draft.
- Matt Vassar conceived and designed the experiments, performed the experiments, contributed reagents/materials/analysis tools, authored or reviewed drafts of the paper, approved the final draft.

### Data Availability
The raw data are provided in Data S1.

### Supplemental Information
Supplemental information for this article can be found online at http://dx.doi.org/10.7717/peerj.4995#supplemental-information.

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
