# Peer review of "Is publication bias present in gastroenterological research? An analysis of abstracts presented at an annual congress"

_PeerJ, doi:10.7717/peerj.4995_

## Round 0.1 · original submission · Minor Revisions

Dear Dr. Meyer,

Your paper has been seen by the editor and two external referees. Both reviewers found the paper to be clear and the findings to be of interest. However, they raised some concerns which need to be considered. If these can be satisfactorily addressed, a revised manuscript is likely to be suitable for publication. Please note that the revised manuscript will undergo a second round of review by the same reviewers. Therefore, acceptance of the manuscript will depend on the completeness of your responses included in the next version of the manuscript. I enclose below the comments received that set out a number of points which will need your attention before we can consider the submission further. I would urge you to give these points your careful attention; in particular, to the statistical issue raised by the reviewer 1.

Regards,

Stefano Menini

·

Basic reporting

In this study, the authors employed a common technique for evaluating publication bias in a specialty, namely exploring the publication outcome of research abstracts in gastroenterology presented at Americas Hepato-Pancreato-Biliary Association Congresses.

The manuscript is very well written, neatly organized and appropriately cited. However, given the straightforward nature of the undertaken study, I found the Introduction to be a little lengthy. For example, the paragraph elaborating on what happens in scientific meetings might be condensed to just 1-2 sentences.

The authors seem to argue that evidence-based evaluations of intervention effectiveness are entirely dependent on the results of analyses which aim to test a specific hypothesis, when (the standard flavor of) null hypothesis significance testing (NHST) in medical research is known to have key limitaitons, not the least of which is a general lack of consideration for treatment effect heterogeneity (both across patients and across providers), preferences, and other factors unrelated to outcome but nonetheless relevant to decision-making.

Please edit any statement about what was done in this study to reflect the past tense.

The Discussion section contains a) items which should be in the Results section; and b) repeated items from the Results section.

Experimental design

The search process for published reports seems comprehensive enough. However, it is safe to assume that some percentage of people contacted by e-mail will simply not respond to a request (regardless of what it is). Given that many authors change titles in their manuscript development (after presentation at a conference), or merge multiple conference abstracts into a single manuscript, the risk of missing publications is not ignorable here. I don't know if I have any good ideas of how to make the search procedure more sensitive, though.

In a similar vein, we should allow the possibility of, for lack of a better term, "methodological evolution", in which the results from the abstract "evolve" into something different due to changes in analytic methods, inclusion criteria, etc.

The description of the logistic regression analysis that was undertaken is vague. A more direct statement such as "Logistic regression was used to evaluate the associations of sample size and study type with publication." Further, were nonlinear relationships with sample size considered? Power has a nonlinear relationship with sample size. Finally, "study type" needs to be clearly defined in the Methods.

Validity of the findings

How did the authors handle the case where abstracts and/or publications may have reported "non-significant" or "NS" test results (without reporting exact P-values)?

Additional comments

That 86% of abstracts which reported a P-value reported statistically significant relationships is concerning. It suggests that the responsibility of "publication bias" lies with researchers and editors alike.

·

Basic reporting

The manuscript has been written in fluent language. Background of knowledge has been given in detail, yet it has made the introduction too long. There are some sentences that should be included and limited to materials & methods section.There are some minor errors in the text to be corrected before final acceptance (such as 'to to' or writing styles of some reference numbers)

Experimental design

Experimental desing has been well constructed and described throughout the methods section. As the authors stated in the discussion section, they have limited number of missed publications via their method of survey.

Validity of the findings

I have seen that prospective studies had higher publication rates than retrospective studies. Is there significance in that finding?

Additional comments

Thanks for delineating and emphasizing the bias present in publishing papers with null or negative results.

---

## Round 0.2 · Minor Revisions

Dear Dr. Meyer,

Thank you for your resubmission. I have now received reports from our reviewers who are generally supportive of publication. However, they suggested minor modifications. Accordingly, I invite you to address the reviewer' s comments and recommendations.

My opinion is that the purpose of the review process should be to improve the manuscript, not just respond to reviewers’ comments. So if the authors agree with the reviewers' suggestions are requested to make the proposed changes. Otherwise, they may challenge reviewers’ thinking in the rebuttal letter.

Regards,

Stefano Menini

·

Basic reporting

The authors commented in their review but not in the manuscript about the shortcomings of using p-values to represent "significant" findings. This has implications to the very concept of "publication bias", and there should be a mention of this in the Discussion.

Experimental design

The authors should incorporate the limitation pointed out in my prior Comment 6 (about non-response to e-mail, changes in title, changes in study results between abstract presentation & publication) as well as within the manuscript. By the way, some of the changes between abstract presentation and final publication are completely appropriate (e.g., refinements to scientific methods in response to peer reviews).

Validity of the findings

The lack of counting "NS" or non-significant results from manuscripts that did not report the value of P-values greater than the chosen significance threshold is a limitation that needs to be reported in the manuscript. Omitting these publications results in a bias in the reported estimate of the incidence of significant results included in the studied publications.

Additional comments

The authors sufficiently responded to my comments, but some of their responses should be directly incorporated into the manuscript itself.

·

Basic reporting

Mentioned before

Experimental design

Mentioned before

Validity of the findings

Mentioned before

Additional comments

In the conclusion section of the abstract, instead of “finding”, it should be “findings”
Readers should take these findings into consideration when reviewing medical literature.

In the statistiscal analysis msection of methodology, you have made revisions about logistic regression and i have been a little bit confused what was intented to mention in the text. Can u please clarify these sentences as simple as possible.


What do you mean with “negative results”? Is it a result of opposite of what had been expected or is it a non-significant result?


In the discussion section:
In our study, the most common study type (removed) design (added) was a retrospective study. Please omit “a” before “retrospective study”.
It has been stated that studies with positive results were more likely to be published earlier than studies with negative or null results. The actual time difference is only 1.5 month. Is is really clinically significantly important?


In the conclusion it has been said that “Abstracts with significant P values had a higher probability of reaching publication.” To say that this should have been put into the logistic regression model, yet it had not been done. So, please make change in logistic regression or change this sentence accordingly.

---

## Round 0.3 · accepted · Accept

Dear Dr. Meyer,

Thank you for submitting a revised version of your manuscript. I am pleased to inform you that your manuscript is accepted for publication in PeerJ in its current form and will now be forwarded to the product editor for copy editing and publication.

I thank all reviewers for their effort in improving the manuscript and the authors for their cooperation throughout the review process

Yours sincerely,

Stefano Menini

·

Basic reporting

No further comments.

Experimental design

No further comments.

Validity of the findings

No further comments.

Additional comments

The authors have successfully addressed - and incorporated into the manuscript - the issues raised in my review.

·

Basic reporting

N/A

Experimental design

N/A

Validity of the findings

N/A

Additional comments

All my recommendations have been well addressed by the authors. Thanks.